# β-function of the level-zero Gross-Neveu model

Dmitri Bykov[*]

- *Steklov Mathematical Institute of Russ. Acad. Sci.,*

  *Gubkina str. 8, 119991 Moscow, Russia*
- *Institute for Theoretical and Mathematical Physics,*

  *Lomonosov Moscow State University, 119991 Moscow, Russia*

**Abstract.** We explain that the supersymmetric $\mathbb{CP}^{n-1}$ sigma model is directly related to the level-zero chiral Gross-Neveu (cGN) model. In particular, beta functions of the two theories should coincide. This is consistent with the one-loop-exactness of the $\mathbb{CP}^{n-1}$ beta function and a conjectured all-loop beta function of cGN models. We perform an explicit four-loop calculation on the cGN side and discuss the renormalization scheme dependence that arises.

*To the memory of A. A. Slavnov, with gratitude*

## Contents

[*]Emails: bykov@mi-ras.ru, dmitri.v.bykov@gmail.com

In our recent work [Byk22b; Byk22c; ABW22; Byk21; Byk22a] we proposed an exact and explicit equivalence between a wide class of integrable sigma models and generalized chiral Gross-Neveu (cGN) models[1]. The principal feature of these sigma models is that the target space is a complex homogeneous space[2]. As the simplest and rather representative example one can consider the $\mathbb{CP}^{n-1}$ sigma model (first formulated in [CJ79; DLDV78; D'A79]).

On the Gross-Neveu side, one considers models with both bosonic and fermionic fields, which is the crucial difference from the traditional purely fermionic Gross-Neveu model [GN74; Wit78]. Another difference from the traditional setup is that the cGN-models in question typically involve auxiliary gauge fields, and part of the gauge symmetry is 'chiral'. As we explained in earlier work [Byk22b; Byk21], a crucial condition is the cancellation of chiral anomalies, which makes it necessary to add fermionic fields to the purely bosonic 'core'. There are, however, many inequivalent ways in which the fermionic degrees of freedom could be added (for example, minimally, supersymmetrically, etc.) [Byk22c].

The fate of the gauge fields in models with vanishing chiral anomalies is rather amusing. It turns out that an admissible (and in many ways best) gauge is in simply setting the gauge fields to zero [Byk22a]. As a result, one arrives at the ungauged version of the cGN-model. In the present paper we will investigate the $\beta$-function of such ungauged model corresponding to the SUSY $\mathbb{CP}^{n-1}$ sigma model. On the one hand, it is well-known that the $\beta$-function of the $\mathbb{CP}^{n-1}$ sigma model is one-loop exact – a result that goes back to [MPS84]. On the other hand, an all-loop $\beta$-function of generalized cGN models was conjectured in [GLM01] (based on earlier results [Kut89]).

---

[1]Chiral Gross-Neveu models are sometimes referred to as non-Abelian Thirring models.

[2]Deformations of models with homogeneous target spaces are also possible and lead to trigonometric and elliptic counterparts of these integrable models. In [Byk22b; Byk21] we studied the RG-flows of such deformed models. However, in the present paper we restrict to the homogeneous (rational) case.

In our special case it is one-loop-exact as well, coinciding the one-loop result of the $\mathbb{CP}^{n-1}$ sigma model. However, at four loops a discrepancy from the conjectured all-loop result has been claimed in [LW03], whose meaning has not been fully elucidated in the past years. As we shall explain below, it can be attributed to a choice of renormalization scheme. Our cGN-based setup, as compared to the abstract setup in [GLM01; LW03], makes the calculation of the $\beta$-function more direct. Since the cGN model is a theory with quartic interactions, one is effectively led to the analysis of divergences that arise in the four-point function. In particular, for generic values of external momenta the four-point function is IR-finite, and the only divergences are in the UV[3].

In the present paper we perform the calculation of the $\beta$-function up to four loops. At two and three loops we find no corrections to the $\beta$-function. At four loops we observe that, in a generic scheme, explicit dependence on regularization appears. We show that the regularization-dependent terms cancel out, if one uses a version of the so-called 'MOM'-scheme [CG79], when the coupling constant is defined as the value of the four-point function for certain (fixed) momenta. In our version one sets two of the four momenta to zero, which is the minimal configuration that ensures IR-finiteness and is still technically simple (for asymmetric versions of MOM-scheme cf. [BL81; CR00]). Although the regularization-dependent terms disappear, in this scheme there is a correction to the $\beta$-function at four loops, proportional to $\zeta(3)$. Interestingly, this type of transcendentality is common for SUSY Kähler sigma models at four loops [GVZ86a; GVZ86b; JJM89], so that the appearance of $\zeta(3)$ seems natural. However, in the case of the $\mathbb{CP}^{n-1}$-model this correction should vanish if one uses a renormalization scheme that preserves $\mathcal{N} = (2,2)$ SUSY, leading to the one-loop-exact $\beta$-function. Although this seems like a paradox at first sight, the discrepancy may again be attributed to a choice of renormalization scheme. The situation is well-known from the theory of the NSVZ[4] $\beta$-function in 4D SUSY theories, where an all-loop $\beta$-function may only be computed in certain special schemes (cf. [KS13; SS22; KS14b; KS14a] and references therein). Below we shall discuss the pros and cons of various schemes.

The structure of the paper is as follows. In section 1 we briefly explain the equivalence between the SUSY $\mathbb{CP}^{n-1}$ sigma model and the gauged chiral Gross-Neveu model.

---

[3]One cannot set all external momenta to zero, though, since this is a special point where IR divergences arise.

[4]'NSVZ' refers to the proposal of Novikov-Shifman-Vainshtein-Zakharov for an exact $\beta$-function in $\mathcal{N} = 1$ SUSY theories in 4D [Shi12].

We also explain that the gauge fields may be completely eliminated by a choice of gauge, which is a peculiar feature of such models. In section 2 we state the conjectured all-loop $\beta$-function [GLM01] for generalized cGN models, viewed as perturbations of CFTs (with affine algebra symmetry) by current-current interactions. Next we recall the history of $\beta$-function calculations in sigma models in section 3, as well as the arguments for its one-loop-exactness in the case of SUSY sigma models with Kähler homogeneous target spaces. We then explain in section 4 that, in the special case of level-zero cGN models, the four-point function, which determines the $\beta$-function, is given solely by crossed-ladder diagrams. Finally, in section 5 we present explicit calculations at three and four loops, commenting on the difference between renormalization schemes.

## 1. The SUSY $\mathbb{CP}^{n-1}$ model as a cGN model

We start by recalling the construction of [Byk22c], where the well-known SUSY $\mathbb{CP}^{n-1}$ sigma model is formulated in a novel way – as a gauged cGN model, with both bosonic and fermionic field content. The fact that supersymmetrization of the $\mathbb{CP}^{n-1}$ model involves coupling it to a fermionic cGN model has been known since the early days of these theories [D'A79], so the novelty here is the realization that the bosonic core is a cGN model in itself, and, moreover, the bosonic and fermionic parts are nontrivially intertwined in a single generalized cGN model. First, we introduce the following fields:

- Bosonic $n$-component vectors: a column-vector $U$ and a row-vector $V$

- Fermionic (Grassmann) $n$-component vectors: a column-vector $C$ and a row-vector $B$

- The above fields are grouped into doublets:

$$\mathscr{U} := \begin{pmatrix} U \\ C \end{pmatrix}, \qquad \mathscr{V} := \begin{pmatrix} V & B \end{pmatrix} \tag{1.1}$$

The worldsheet is assumed to be[5] $\mathbb{C} \simeq \mathbb{R}^2$, with complex coordinate $z = x + i\,y$. In

---

[5]Generalizations to Riemann surfaces are possible [Byk22a] but will not be discussed here.

terms of the fields introduced above we write down the following Lagrangian:

$$\mathscr{L}_{\mathbb{CP}} = 2\left(\mathscr{V}\cdot\overline{\mathcal{D}}\mathscr{U} + \overline{\mathscr{U}}\cdot\mathcal{D}\overline{\mathscr{V}}\right) + \frac{\varkappa}{2\pi}\,\mathrm{Tr}(J\overline{J})\,, \tag{1.2}$$

where the covariant derivative is defined as follows:

$$\overline{\mathcal{D}} = \frac{\partial}{\partial\overline{z}} + i\,\overline{\mathcal{A}}_{\mathrm{super}}\,, \qquad \overline{\mathcal{A}}_{\mathrm{super}} = \begin{pmatrix}\overline{\mathcal{A}} & 0 \\ \overline{\mathcal{W}} & \overline{\mathcal{A}}\end{pmatrix} \tag{1.3}$$

The corresponding gauge group is a triangular subgroup of $\mathrm{SL}(1|1)$, isomorphic to $\mathbb{C}^* \ltimes \mathbb{C}_f$, where $\mathbb{C}_f$ is a Grassmann one-dimensional vector space. In particular, in (1.3) $\mathcal{A}$ is a bosonic gauge field, and $\mathcal{W}$ a fermionic one. The variable $J$ in the interaction term in (1.2) is the so-called 'moment map', or more simply a current, defined as follows:

$$\frac{1}{2\pi}\,J := U\otimes V - C\otimes B \quad \in \mathfrak{gl}_n\,, \tag{1.4}$$

whereas $\overline{J}$ is its Hermitian conjugate. Finally, $\varkappa$ is the coupling constant in the Lagrangian (1.2). To conclude with the notations, in quantum theory the Boltzmann weight in the path integrals is defined to be $e^{-S}$ with the action $S = \int d^2z\,\mathscr{L}_{\mathbb{CP}}$, where $d^2z := dx \wedge dy = \frac{i}{2}dz \wedge d\overline{z}$.

The Lagrangian (1.2) is an example of a chiral Gross-Neveu model. The first two terms are first-order kinetic terms, whereas the interaction is a quartic coupling, just like in the cGN model. An explicit rewriting of (1.2) as a cGN Lagrangian is possible, if one introduces the Dirac doublets $\Psi = \begin{pmatrix}U \\ \overline{V}\end{pmatrix}$ (a bosonic one) and $\Theta = \begin{pmatrix}C \\ \overline{B}\end{pmatrix}$ (a fermionic one). The main difference is that in the original cGN model the fields are fermionic, whereas here we have both bosonic and fermionic fields.

As mentioned above, the gauge group in question is $\mathbb{C}^* \ltimes \mathbb{C}_f$. One can show that an admissible (partial) gauge is $\overline{U}U = 1$, $\overline{C}U = 0$. The latter (fermionic) condition fully fixes the $\mathbb{C}_f$ part of the gauge symmetry, whereas the former reduces $\mathbb{C}^*$ down to $\mathrm{U}(1)$ – the gauge symmetry typical of the standard formulation of the $\mathbb{CP}^{n-1}$ sigma model. Upon imposing this gauge, one eliminates the fields $V, \overline{V}$ from (1.2), which can be easily done, since these fields enter quadratically and effectively without derivatives. The resulting Lagrangian corresponds to the standard form of the SUSY $\mathbb{CP}^{n-1}$ model.

All these steps were discussed in detail in [Byk22c], but to match with standard notations and to familiarize the reader with our formalism let us show how to proceed to the geometric form of the sigma model in the purely bosonic case. To this end, we set $B = C = 0$ and 'integrate out' $V, \bar{V}$ from the Lagrangian (1.2). As a result, we get the action

$$S = \frac{1}{2\pi \varkappa} \int d^2 z \, \frac{4|\overline{\mathcal{D}}U|^2}{\overline{U}U} = \{ \text{ choosing the gauge } \overline{U}U = 1 \} = \tag{1.5}$$

$$= \frac{1}{2\pi \varkappa} \int d^2 z \, \mathcal{D}_\alpha U \mathcal{D}_\alpha \bar{U} + \frac{i}{2\pi \varkappa} \int dU \wedge d\bar{U} \,,$$

where $\partial_\alpha = \left( \frac{\partial}{\partial x}, \frac{\partial}{\partial y} \right)$. The above action corresponds to the standard $\mathbb{CP}^{n-1}$-model with a topological term (cf. [MPS84]). In particular, the factors of $2\pi$ in (1.2) and (1.4) lead to the conventional factor of $\frac{1}{2\pi}$ in front of the sigma model action in geometric form.

## 1.1. Eliminating the gauge field by choice of gauge

As we just discussed, $\overline{U} \cdot U = 1, \overline{C} \cdot U = 0$ is an admissible partial gauge for the SUSY model. The virtue of the cGN formulation (1.2) is that there is a different gauge, which fully fixes the gauge symmetry and is more convenient in many ways.

To formulate this gauge, and for future use, we introduce an auxiliary field $\mathcal{B}$ and perform a quadratic transformation on the system (1.2):

$$\mathscr{L} = 2 \left( \mathscr{V} \cdot \overline{\mathcal{D}}\mathscr{U} + \overline{\mathscr{U}} \cdot \mathcal{D}\overline{\mathscr{V}} \right) + \frac{1}{2\pi} \left( \mathrm{Tr} \left( \mathcal{B}\overline{\mathcal{B}} \right) + i \, \varkappa^{\frac{1}{2}} \mathrm{Tr} \left( J\overline{\mathcal{B}} \right) + i \, \varkappa^{\frac{1}{2}} \mathrm{Tr} \left( \overline{J}\mathcal{B} \right) \right) \tag{1.6}$$

The quadratic (Hubbard-Stratonovich) transformation trick is standard for cGN models and has been used in $\beta$-function calculations in [Des88; Bon$^+$90].

Next we decompose $\mathcal{B} = \mathcal{B}_0 \cdot \mathbf{1}_n + \mathcal{B}_\perp$, where $\mathcal{B}_0 = \frac{1}{n}\mathrm{Tr}(\mathcal{B})$ and $\mathcal{B}_\perp$ is the $\mathfrak{sl}_n$ (traceless) part of $\mathcal{B}$. The gauge is then

$$\mathcal{A} = \frac{1}{2}\varkappa^{\frac{1}{2}} \mathcal{B}_0 \,, \qquad \mathcal{W} = 0 \,. \tag{1.7}$$

As a result, the gauge field is completely eliminated [Byk22a], and one is effectively

left with the simple part $\mathcal{B}_\perp \in \mathfrak{sl}_n$ in the interaction terms[6].

Let us explain why the choice (1.7) is possible. Gauge transformations of $\bar{\mathcal{A}}_{\text{super}}$ in (1.3) are the standard ones:

$$i\bar{\mathcal{A}}_{\text{super}} \mapsto g^{-1}\left(i\bar{\mathcal{A}}_{\text{super}}\right)g + g^{-1}\bar{\partial}g\,, \qquad g = \begin{pmatrix} e^\zeta & 0 \\ \chi & e^\zeta \end{pmatrix} \qquad (1.8)$$

In components, this reads

$$i\bar{\mathcal{A}} \mapsto i\bar{\mathcal{A}} + \bar{\partial}\zeta \qquad (1.9)$$

$$i\bar{\mathcal{W}} \mapsto i\bar{\mathcal{W}} + e^{-\zeta}\left(\bar{\partial}\chi - \bar{\partial}\zeta \cdot \chi\right)\,. \qquad (1.10)$$

The gauge fixing is achieved in two steps. First, one performs a gauge transformation with $\chi = 0$, choosing $\zeta$ so that the gauge-transformed field $\bar{\mathcal{A}}$ satisfies (1.7), i.e. $i(\bar{\mathcal{A}} - \frac{1}{2}\varkappa^{\frac{1}{2}}\bar{\mathcal{B}}_0) + \bar{\partial}\zeta = 0$. The solution is given by the Cauchy-Green formula

$$\zeta(z,\bar{z}) = -\frac{i}{\pi}\int d^2w\, \frac{\bar{\mathcal{A}}(w,\overline{w}) - \frac{1}{2}\varkappa^{\frac{1}{2}}\bar{\mathcal{B}}_0(w,\overline{w})}{z - w}\,. \qquad (1.11)$$

Here one assumes that the decay conditions on the fields at infinity are such that the integral makes sense. One then performs a second gauge transformation with $\zeta = 0$, choosing $\chi$ in such a way that $i\bar{\mathcal{W}} + \bar{\partial}\chi = 0$, which again relies on the same Cauchy-Green formula. As a result, the gauge (1.7) is imposed.

**1.2. The ungauged cGN model** In the gauge (1.7) the model (1.2) simplifies as follows:

$$\mathcal{L}_{\mathbb{CP}} = 2\left(\mathcal{V}\cdot\bar{\partial}\mathcal{U} + \overline{\mathcal{U}}\cdot\partial\overline{\mathcal{V}}\right) + \frac{\varkappa}{2\pi}\,\text{Tr}(J_\perp\overline{J}_\perp)\,, \qquad (1.12)$$

where $J_\perp$ is the traceless part of $J$. This is obtained from (1.6) upon integrating out $\mathcal{B}$, assuming $\mathcal{B} \in \mathfrak{sl}_n$. From now on we simply write $J$ in place of $J_\perp$, keeping in mind that we are considering $\mathfrak{sl}_n$ in place of $\mathfrak{gl}_n$ as the relevant symmetry algebra.

---

[6]One could as well choose the gauge $\mathcal{A} = 0$, as in [Byk22a]. In that case the interaction term would split as $\varkappa^{\frac{1}{2}}\text{Tr}(\bar{J}\mathcal{B}) = \varkappa^{\frac{1}{2}}\text{Tr}(\bar{J}\mathcal{B}_\perp) + \varkappa^{\frac{1}{2}}\text{Tr}(\bar{J})\mathcal{B}_0$. The coupling constant in front of the first (traceless) term undergoes renormalization, as described below, whereas the one in front of the trace part is not renormalized (cf. [Byk22b]). It is to avoid dealing with these two different terms that we have chosen the gauge (1.7).

The kinetic terms in (1.12) represent what is known as a $\beta\gamma$-system [Wit07; Nek05]. For $\varkappa = 0$ this is a CFT with a left/right Kac-Moody symmetry. Moreover, in the absence of interactions, the corresponding symmetry group is $\widehat{\mathrm{GL}(n|n, \mathbb{C})}$ with the natural transformations

$$\mathscr{U} \mapsto h(z)\,\mathscr{U}\,, \qquad \mathscr{V} \mapsto \mathscr{V}h(z)^{-1}\,, \qquad h(z) \in \mathrm{GL}(n|n, \mathbb{C})\,. \tag{1.13}$$

The interaction term breaks this huge symmetry. From the point of view of the free system the variable $J$ featuring in the interaction term is nothing but the Kac-Moody current of $\widehat{\mathrm{SL}(n, \mathbb{C})} \subset \widehat{\mathrm{GL}(n|n, \mathbb{C})}$, where the diagonal embedding is assumed. For $\varkappa = 0$, the current is holomorphic, $\bar{\partial}J = 0$, whereas for non-zero $\varkappa$ this condition is replaced by $\bar{\partial}J = \frac{\varkappa}{2}[J, \bar{J}]$. The latter equation implies that the current $J dz + \bar{J} d\bar{z}$ is both conserved and flat, signalling potential integrability of the model.

In other words, our system may be seen as a concrete realization of a more general setup, where one considers a Lagrangian of the form

$$\mathscr{L} = \mathscr{L}_{\mathrm{CFT}} + \frac{\varkappa}{2\pi} \operatorname{Tr}(J\bar{J})\,, \tag{1.14}$$

which is a current-current perturbation of a conformal field theory with a Kac-Moody symmetry (in our case the CFT being a free system). Such systems have been studied in the past, in particular in [FOZ93; MQS08] in relation to sigma models.

## 2.   The conjectured $\beta$-function

Given the system (1.12), or more generally (1.14), a natural task is in the computation of the $\beta$-function of $\varkappa$. In the purely fermionic case of the traditional cGN model this question was addressed long ago[7]: the one-loop result in [GN74], where asymptotic freedom of the model was established, the two-loop result in [Des88; Bon$^+$90] and even a three-loop result in [BG99].

A natural generalization is in considering the system (1.14) with more general field

---

[7]In the case of the non-chiral GN model there are more results: we refer to [LR91; Gra91] for the three-loop case and to [GLS16] for the most recent (four-loop) results (see references therein for earlier work).

content. A scheme for the computation of the $\beta$-function of such cGN-models was proposed as early as in [Kut89]. It was argued that the only ingredient necessary for developing the perturbation theory is the OPE of the chiral currents, which takes the well-known form

$$J^a(z)J^b(w) = \frac{k\,\delta^{ab}}{(z-w)^2} + \frac{i\,f^{ab}_c\,J^c(w)}{z-w} + \dots \,, \tag{2.1}$$

where $J^a = \mathrm{Tr}(J\tau^a)$ are the components of the current, and $k$ is the level. Here $\tau^a$ are the unit-normalized Hermitian generators of the corresponding (simple) Lie algebra $(\mathrm{Tr}(\tau^a\tau^b) = \delta^{ab})$, and $f^{ab}_c$ are its structure constants defined by $[\tau^a, \tau^b] = i\,f^{ab}_c\,\tau^c$. In our applications the Lie algebra is $\mathfrak{sl}_n$, and the current is the traceless part $J_\perp$ of (1.4), as already discussed.

The abstract theory defined by (1.14) and (2.1) has two parameters: $\varkappa$ and $k$. The approach of [Kut89] relied on a perturbation theory in $\frac{1}{k}$, with $\lambda := \varkappa k$ fixed (the latter could be viewed as a 't Hooft coupling of sorts), and the calculation was carried out to leading order. In [GLM01] the authors pushed the method further and conjectured an all-loop $\beta$-function, valid for all values of $\varkappa$ and $k$. Moreover, their result applies to very general systems of the type (1.14), even in the case of several couplings (i.e. multiple current-current deformations). For the case of a single coupling $\varkappa$ as in (1.14), assuming the current algebra (2.1), their result reads:

$$\beta_\varkappa = -\frac{\mathsf{C}_2\,\varkappa^2}{\left(1 + \frac{1}{2}k\,\varkappa\right)^2}\,, \tag{2.2}$$

where $\mathsf{C}_2$ is the value of the quadratic Casimir of the symmetry algebra, defined by $\frac{1}{2}f_{abc}f_{abd} = \mathsf{C}_2\delta_{cd}$ ($\mathsf{C}_2 = n$ in the case of $\mathfrak{sl}_n$).

To apply the above formula to our system (1.12) what remains is to compute the level $k$. Taking into account the expression (1.4) for the Kac-Moody current and the elementary correlators $\langle U_i(z)V_j(w)\rangle = \langle B_i(z)C_j(w)\rangle = \frac{\delta_{ij}}{z-w}$, one easily finds

$$\langle J_{ij}(z)J_{i'j'}(w)\rangle_{\varkappa=0} = 0\,, \tag{2.3}$$

as the contributions of bosons and fermions cancel exactly. In other words, the level

vanishes, implying a one-loop-exact $\beta$-function:

$$k = 0 \qquad \text{so that} \qquad \beta_{\varkappa} = -n\,\varkappa^2 \qquad (2.4)$$

On the one hand, this is expected from the standpoint of the $\mathbb{CP}^{n-1}$ sigma model, since it is known that for *Hermitian symmetric* target spaces[8] (and even more generally for Kähler homogeneous spaces admitting a GLSM description, as we explain below) the $\beta$-function is one-loop exact in the SUSY case. This was found in [MPS84] based on NSVZ-type (instanton-related) arguments. In these terms the one-loop-exactness of the $\beta$-function is a 2D analogue of the analogous phenomenon in $\mathcal{N} = 2$ theories in 4D. For completeness, we should mention that, in the past years, an even more complete parallel with 4D NSVZ results has been found for models with $(0,2)$ SUSY [CS12; Che+14; CS19], see [Shi18] for a review. We expect that our cGN formalism could be extended to models with $(0,2)$ SUSY, so that those models would be amenable to a similar analysis.

On the other hand, in an attempt to check the conjectured result (2.2) for finite values of $k$ the authors of [LW03] considered the extreme case of $k = 0$. This value was chosen since 1) it is polar to $k = \infty$ and 2) simplifies the calculations as there is only one non-zero term in the OPE (2.1). As we have just seen, however, there are very concrete systems (SUSY sigma models) that realize this abstract setup. The claim of [LW03] was that an explicit calculation in a hard (coordinate space) cutoff scheme leads to a correction $\beta^{(4)} \overset{?}{=} -\frac{\pi}{160}(6 + \pi^2)n^2\varkappa^5$ to the exact formula at four loops. The status of this result has not been clarified to present day (cf. [BL21]). One should note that it is not in contradiction with the conjectured form (2.4), since the $\beta$-function depends on the renormalization prescription (scheme). Indeed, upon a change of variables $\varkappa = \varkappa(\widehat{\varkappa})$ the RG equation $\dot{\varkappa} = \beta(\varkappa)$ is transformed into $\dot{\widehat{\varkappa}} = \widehat{\beta}(\widehat{\varkappa})$, where

$$\widehat{\beta}(\widehat{\varkappa}) = \left(\frac{\partial \varkappa}{\partial \widehat{\varkappa}}\right)^{-1} \beta(\varkappa(\widehat{\varkappa})) \qquad (2.5)$$

Consider now a $\beta$-function whose only contributions are at one, four and possibly higher loops (the setup relevant for our application): $\beta(\varkappa) = \beta^{(1)}\varkappa^2 + \beta^{(4)}\varkappa^5 + \ldots$, and

---

[8]At least of classical groups

a corresponding change of variables $\varkappa = \widehat{\varkappa} + c\widehat{\varkappa}^4 + \ldots$. The transformed $\beta$-function is then

$$\widehat{\beta}(\widehat{\varkappa}) = \beta^{(1)}\widehat{\varkappa}^2 + (\beta^{(4)} - 2c\beta^{(1)})\widehat{\varkappa}^5 + \ldots \qquad (2.6)$$

Thus, picking $c = \frac{\beta^{(4)}}{2\beta^{(1)}}$, one can cancel the unwanted contribution at four loops.

The above ambiguities are well-known in the context of the NSVZ $\beta$-function in 4D theories [JJN96; JJN97], making the latter notoriously hard to prove explicitly: cf. [KS13; SS22] for the abelian case, as well as [KS14b; KS14a] for an explicit comparison of different regularization and renormalization prescriptions. The conjectured $\beta$-function (2.2) should be seen as a 2D analogue, albeit applicable in a wider context, beyond the scope of SUSY models. One should therefore expect similar difficulties related to scheme dependence.

Below we present an explicit four-loop computation of the $\beta$-function using a method that allows (partially) keeping track of the regularization dependence. We find that the dependence on regularization may be cancelled by a choice of renormalization scheme, as in (2.6). Besides, choosing appropriately the value of $c$, one can cancel the four-loop contribution to the $\beta$-function altogether (this is essentially the line of thinking pioneered in [JJN96; JJN97] in the context of the NSVZ $\beta$-function). Finding a natural regularization/renormalization scheme that would lead to the exact $\beta$-function remains a challenge for the future.

## 3.   History of sigma model calculations

Prior to our calculation, let us review the argument that the $\beta$-function of the $\mathbb{CP}^{n-1}$ model is one-loop-exact, recalling some key results. Calculation of $\beta$-functions for sigma models has a long history, dating back to the two-loop result of [Fri80]. The four-loop result for purely bosonic sigma models can be found in [JJM89] (for more references see [Ket09]). Since we are mostly interested in the SUSY setup, here the two-loop result was obtained in [AGFM81] and the four-loop result in [GVZ86a; GVZ86b], all of them using the $\overline{\text{MS}}$ scheme. The latter reads:

$$\beta^{(4)}_{i\bar{j}} \sim \zeta(3) \cdot \partial_i \bar{\partial}_j \, \Delta K, \qquad \Delta K = R_{\kappa\bar{\lambda}\mu\bar{\nu}}R^{\sigma\bar{\lambda}\tau\bar{\nu}}R_{\sigma}{}^{\kappa}{}_{\tau}{}^{\mu} - R_{\kappa\bar{\lambda}\mu\bar{\nu}}R^{\mu\bar{\nu}\sigma\bar{\tau}}R_{\sigma\bar{\tau}}{}^{\kappa\bar{\lambda}} \qquad (3.1)$$

The notation $\Delta K$ means that this could be viewed as a correction to the Kähler potential, since, as we recall, models with $(2, 2)$ SUSY correspond to Kähler target spaces. The corresponding result for purely bosonic models is more complicated and involves, apart from $\zeta(3)$, additional rational coefficients (such as $a + b\,\zeta(3)$, where $a$ and $b$ are rational). Moreover, it was observed in [JJM89] that the terms proportional to $\zeta(3)$ exactly coincide with the SUSY result. The five-loop contribution to the $\beta$-function was calculated in [GKZ87], where it was shown that it can be cancelled by a redefinition of the metric (i.e., a change of scheme).

As mentioned earlier, there is also the result that for *Hermitian symmetric* target spaces the $\beta$-function is one-loop exact. This was originally proposed in [MPS84], but there are at least two other ways to arrive at this conclusion and even to generalize it, which we now recall.

First, a direct superspace analysis of the admissible counterterms in [HPS86] has shown that, at higher loops (with the exception of the one-loop case), the $\beta$-function is always of the form $\beta^{(l)}_{i\bar{j}} \sim \partial_i \bar{\partial}_j \, \Delta K^{(l)}$ ($l \geqslant 2$ is the number of loops), where $\Delta K^{(l)}$ is a globally defined function on the manifold. Speaking more formally, the correction is cohomologically trivial. The function $\Delta K^{(l)}$ is a scalar built out of the Riemann tensor and its covariant derivates, as in the four-loop example (3.1). If we now restrict to a homogeneous (not even necessarily symmetric) target space with an invariant metric, the corresponding $\Delta K^{(l)}$-functions would have to be invariant as well and, as such, they can only be constants. Thus, all higher-loop corrections to the $\beta$-functions of sigma model with Kähler homogeneous target spaces vanish, if one uses a renormalization scheme that manifestly preserves $\mathcal{N} = (2, 2)$ SUSY.

**3.1.   GLSM-based arguments** Another take on this result may be obtained by noting that Kähler homogeneous spaces admit gauged linear sigma model (GLSM) formulations. The method, introduced in [D'A$^+$83] (and used many times thereafter, for example in [Wit93; MP95; NS09]), consists in integrating out chiral matter fields and constructing an effective action for the gauge superfield. Recall the superfield form of the SUSY action:

$$S = \int d^2 z \int d^4\theta \left( \sum_{j=1}^{n} \overline{\Phi}_j e^{-V} \Phi_j + \frac{\xi}{2\pi} V \right) + \frac{\theta}{2\pi} \int d^2 z \, F_{z\bar{z}} , \qquad (3.2)$$

where $V$ is the gauge superfield, $\xi$ the FI term and $\theta$ the topological $\theta$-angle. The parameter $\xi$ is proportional to the squared radius of the target space and is related to our parameter $\varkappa$ as $\xi = \frac{1}{\varkappa}$ (compare (1.5) with the Lagrangian (3.4) below).

One way to see that (3.2) really leads to the $\mathbb{CP}^{n-1}$ sigma model is to exclude the gauge field $V$ using its e.o.m.:

$$V = \log\left(\frac{2\pi}{\xi}\sum_{j=1}^{n}\overline{\Phi}_j\Phi_j\right). \tag{3.3}$$

Substituting back into the action, one gets $\frac{\xi}{2\pi}V$ (up to a constant), which is the Kähler potential of $\mathbb{CP}^{n-1}$. The form (3.2) is useful, though, as the 'matter' fields $\Phi_j$ enter quadratically and therefore may be easily integrated out. To start with, one may rewrite (3.2) in components [D'A+83]:

$$
\begin{aligned}
\mathscr{L} = & \sum_{j=1}^{n}\overline{U}_j\left(-(\partial - iA)_\mu(\partial - iA)^\mu + \bar{\sigma}\sigma - D\right)U_j + \\
& + \sum_{j=1}^{n}\overline{\psi}_j\left(\gamma^\mu(\partial - iA)_\mu + \frac{1}{2}(1 + i\gamma_5)\bar{\sigma} + \frac{1}{2}(1 - i\gamma_5)\sigma\right)\psi_j - \\
& - \sum_{j=1}^{n}\overline{F}_jF_j + \bar{\chi}\left(\sum_{j=1}^{n}\overline{U}_j\psi_j\right) + \left(\sum_{j=1}^{n}U_j\overline{\psi}_j\right)\chi + \frac{1}{2\pi}\left(\xi D + \theta F_{z\bar{z}}\right).
\end{aligned}
\tag{3.4}
$$

This Lagrangian deserves some comments. First, $(U_j, \psi_j, F_j)$ belong to the chiral multiplets, whereas $(A_\mu, \sigma, \bar{\sigma}, \chi, \overline{\chi}, D)$ belong to the vector multiplet. $\chi$ and $\psi$ are fermions, and all other fields are bosonic. We have not included a dynamical term for the gauge field, which means that the vector multiplet is auxiliary and all of its components enter as auxiliary fields. For example, $D$ acts as a Lagrange multiplier for the constraint $\sum_{j=1}^{n}\overline{U}_jU_j = \frac{\xi}{2\pi}$. Imposing this constraint and integrating out $\sigma, \bar{\sigma}$ would lead to the quartic coupling $\left(\overline{\psi}\frac{1+i\gamma_5}{2}\psi\right) \times \left(\overline{\psi}\frac{1-i\gamma_5}{2}\psi\right)$ of the fermionic cGN model.

We return to the $\beta$-function. A major point is that the FI term in (3.4), together with the $\theta$-term, may be rewritten as [Wit93]

$$\mathscr{L}_{\mathrm{FI}} = \frac{1}{2\pi}\left(\xi D + \theta F_{z\bar{z}}\right) = \mathsf{Re}\left(\frac{1}{2\pi}\int d^2\theta\, t\, \Sigma\right), \tag{3.5}$$

where $t = \xi - i\theta$ is the 'complexified FI parameter' and $\Sigma$ a twisted chiral superfield encoding the gauge field strength $F_{z\bar{z}}$ and the scalar fields $\sigma$ and $D$. In the above formula, $\frac{t}{2\pi}\Sigma$ is the tree-level value of the twisted superpotential $\widetilde{W}(\Sigma)$. Allowing a general twisted superpotential leads, in components, to the following Lagrangian:

$$\mathsf{Re}\left(\int d^2\theta\, \widetilde{W}(\Sigma)\right) = \mathsf{Re}\left((D + iF_{z\bar{z}})\frac{\partial \widetilde{W}(\sigma)}{\partial \sigma}\right) + \text{fermions} \qquad (3.6)$$

In particular, the coupling of $D$ to $\sigma$ has the form $D \cdot \left(\frac{\partial \widetilde{W}(\sigma)}{\partial \sigma} + \text{c.c.}\right)$, i.e. it involves a sum of holomorphic and anti-holomorphic functions. The effective twisted superpotential may be extracted from this coupling.

In doing a background field calculation of $\widetilde{W}$ with constant background values of $D$ and $\sigma$, at one loop the only graph that contributes is shown in Fig. 1. The corresponding contribution to the effective action is ($\Lambda \gg |\sigma|$ is the cutoff)

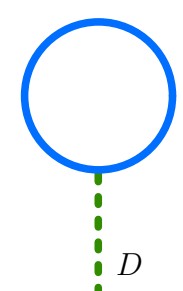

$$S_{\text{eff}}^{(1\text{-loop})} = D \cdot \left(\frac{n}{(2\pi)^2}\int \frac{d^2p}{p^2 + |\sigma|^2}\right) =$$
$$= \frac{n}{4\pi} D \log\left(\frac{\Lambda^2}{|\sigma|^2}\right), \qquad (3.7)$$

Figure 1: Diagram contributing to the renormalization of $\xi$. The blue line denotes the 'matter' fields $U, \bar{U}$.

which is a sum of holomorphic and anti-holomorphic function, as required. The twisted superpotential, at one loop, is thus

$$\widetilde{W}(\Sigma) = \frac{1}{2\pi}\left(t + n\, \log\left(\frac{\Sigma}{\Lambda}\right)\right)\cdot \Sigma \qquad (3.8)$$

To get rid of the cutoff, we renormalize the coupling constant $\xi$ ($\theta$ remains unchanged):

$$\xi = \xi_{\text{R}} + n \cdot \log\left(\frac{\Lambda}{\mu}\right) \qquad (3.9)$$

The shift (3.9) corresponds to the one-loop $\beta$-function. As for higher loops, these would necessarily involve propagators of the gauge field and its superpartners, since

matter fields enter purely quadratically. A typical two-loop diagram is shown in Fig. 2. Here one runs into trouble, as the Lagrangian (3.4) involves no quadratic piece for the gauge field and, hence, the propagators are undefined. To cure this situation, one solution is to embed the $\mathbb{CP}^{n-1}$-model in a *linear* sigma model by adding a term $\frac{1}{e^2} \int d^4\theta \, \Sigma\bar{\Sigma}$ to the Lagrangian, which leads to the standard kinetic term $\frac{1}{e^2} F_{z\bar{z}}^2$ for the gauge field. In two dimensions, the coupling $e$ has dimension of mass, meaning that a diagram with gauge field propagators would be suppressed by powers of mass. For example, the diagram in Fig. 2 is proportional to $\frac{e^2}{|\sigma|^2}$, which is clearly not a sum of holomorphic and anti-holomorphic function.

Thus, supersymmetry requires that, once all relevant diagrams are included, any contributions to the effective twisted superpotential involving powers of $e$ should vanish. As a result, the twisted superpotential is independent of $e$, one-loop exact and given by (3.8). Consequently, the renormalization of $\xi$ is one-loop-exact as well.

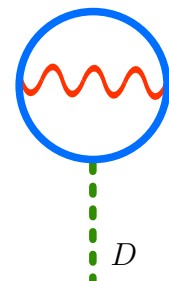

Figure 2: Diagram with gauge field insertion, proportional to $\frac{e^2}{|\sigma|^2}$.

To summarize, from the modern perspective the one-loop exactness of the $\beta$-function follows from the existence of the GLSM formulation, since in this case all couplings are encoded in the FI terms that receive radiative corrections at one loop only.

## 4.   The $\beta$-function from crossed ladder diagrams

Having reviewed the standard methods for studying sigma model $\beta$-functions, we pass to our alternative formulation (1.12) in terms of a cGN model. In fact, we will be mostly using its equivalent version (1.6) with an auxiliary field $\mathcal{B}$. To study the $\beta$-function of this model we impose the gauge (1.7) and rewrite (1.6) as follows:

$$\mathscr{L} = 2 \left( V\overline{D}U + B\overline{D}C + \overline{U}DV + \overline{C}DB \right) + \frac{1}{2\pi} \mathrm{Tr} \left( \mathcal{B}\overline{\mathcal{B}} \right) , \tag{4.1}$$

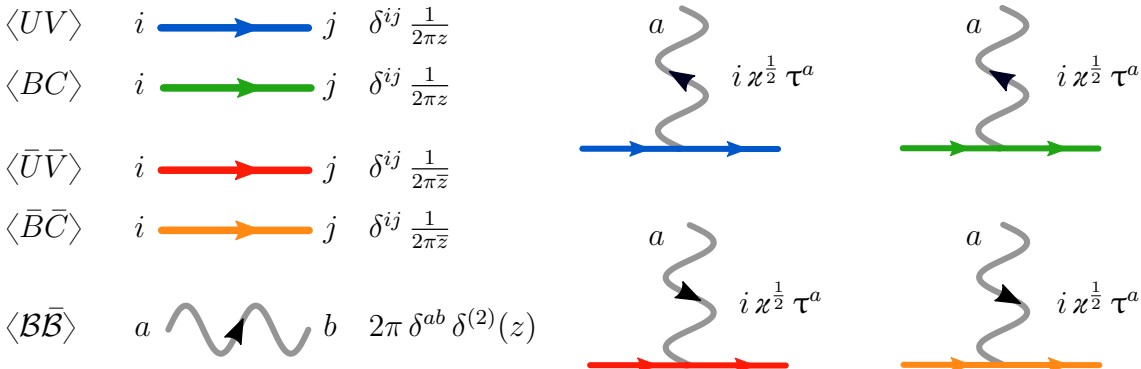

Figure 3: Feynman rules of the model (4.1). We have expanded $\mathcal{B} = \sum \mathcal{B}_a \tau^a$, where $\mathcal{B}_a$ are complex numbers, and the generators $\tau^a \in \mathfrak{sl}_n$ are Hermitian and unit-normalized: $\mathrm{Tr}(\tau^a \tau^b) = \delta^{ab}$.

where we now assume $\mathcal{B} \in \mathfrak{sl}_n$. Here we used the explicit form (1.4) of the current, and defined a *new* 'covariant derivative' as

$$\overline{D}U = \overline{\partial}U + \frac{i}{2}\varkappa^{\frac{1}{2}}\overline{\mathcal{B}}U. \tag{4.2}$$

In Fig. 3 we list the Feynman rules of this theory.

The auxiliary field $\mathcal{B}, \bar{\mathcal{B}}$ entering the covariant derivatives is only formally a 'gauge field', since the term $\mathrm{Tr}\left(\mathcal{B}\overline{\mathcal{B}}\right)$ in the above Lagrangian clearly violates gauge invariance. We should also emphasize that the introduction of this field is merely a technical tool for the simplification of combinatorics in Feynman diagrams corresponding to the Green's functions of the fundamental fields $U, V, B, C$. If one additionally wants to allow the $\mathcal{B}, \bar{\mathcal{B}}$ fields in the external lines, one would have to work in a two-coupling formalism of [LR91][9], adding an independent bare quartic coupling to (4.1).

In order to compute renormalization of the coupling constant $\varkappa$ we will consider the four-point function

$$\frac{1}{16}G_4^{iji'j'}(p_1, \cdots, p_4) := \int \prod_{j=1}^{4} d^2 w_j \, e^{i(p_j, w_j)} \langle U^i(w_1)V^j(w_2)\bar{U}^{i'}(w_3)\bar{V}^{j'}(w_4)\rangle \tag{4.3}$$

---

[9]The paper [LR91] deals with the non-chiral GN model. The field $\sigma$ used therein to split the quartic vertex is an SU($n$) singlet and is akin to the field $\sigma$ of (3.4). This type of splitting is used in the large-$n$ analysis, both in the chiral and non-chiral case [GN74; Wit78].

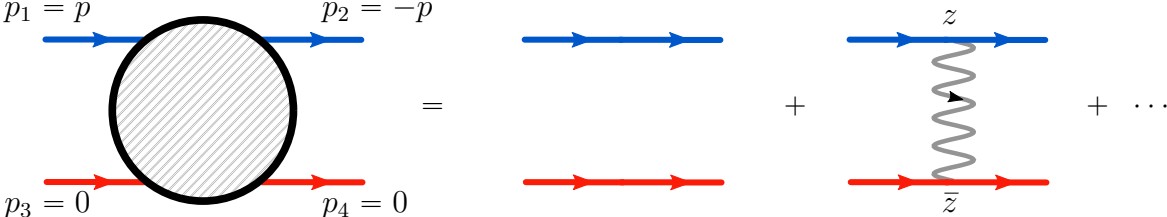

Figure 4: The correlation function $G_4(p_1, \cdots, p_4)$.

with external lines taken in momentum representation, and the scalar product is defined as[10] $(p, w) := pw + \bar{p}\bar{w}$. As shown in Fig. 4, to leading order we get the (connected) contribution

$$- (2\pi)\, \varkappa \sum_a (\tau_a)^{ij} (\tau_a)^{i'j'} \times \frac{1}{\bar{p}_1 \cdot \bar{p}_2 \cdot p_3 \cdot p_4} \times \delta^{(2)} \left( \sum_{j=1}^4 p_j \right) \tag{4.4}$$

For brevity we will strip off the indices and amputate the overall momentum-dependent factor, denoting the resulting contribution to the 4-point function by $\widehat{G}_4(p_1, \cdots, p_4)$. At tree level one has

$$-\frac{1}{2\pi} \widehat{G}_4^{\text{tree}} = \varkappa \sum_a \tau_a \otimes \tau_a \tag{4.5}$$

As the next step, we will simplify the kinematics by setting $p_3 = p_4 = 0$, which implies $p_2 = -p_1 := p$. Despite the fact that the four-point function $G_4(p_1, \cdots, p_4)$ is singular in this limit due to the poles of the external legs, as in (4.4), the amputated function $\widehat{G}_4(p, -p, 0, 0) := \widehat{G}_4(p)$ is regular. As we shall see, the non-zero momentum $p \neq 0$ serves as a natural infrared regulator (this special kinematics has been taken advantage of, in a similar context, as early as in [VV97]).

A curious observation is that the diagrams that contribute to the four-point function (4.3) of bosonic fields involve *exclusively* the (crossed) ladders of the bosonic fields themselves. This is not only true at tree level (Fig. 4) but at all higher loops as well (cf. Figs. 6 and 7 below). Fermions could appear in loops with gauge fields emanating at the nodes, however one would as well have diagrams with bosonic fields propagating in these same loops, and the two contributions would cancel exactly (see Fig. 5). This is the specific situation corresponding to level $k = 0$, which makes it different from the purely bosonic or purely fermionic Gross-Neveu models, where, on top of the ladder

---

[10]We use the same symbol for a 2-vector and for its holomorphic component. Whenever a scalar product of 2-vectors is implied, round brackets are included.

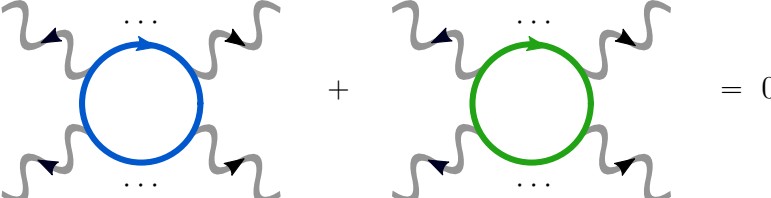

Figure 5: Cancellation of matter loop diagrams in the $\mathsf{k} = 0$ model.

diagrams, one would additionally have those non-vanishing loop insertions.

The same argument implies that the two-point function $\langle U(w_1)V(w_2)\rangle$ receives no quantum corrections, and therefore no renormalization of the field is required.

**4.1. One loop** A one-loop calculation is a good starting point to illustrate our technique. Although we have taken the external lines in momentum space, perturbation theory will be constructed using coordinate space Feynman rules, as shown in Fig. 3. At one loop there are two diagrams, shown in Fig. 6. Let us first take a look at the left one, whose contribution is

$$\text{Fig. 6 left} = \varkappa^2 \, \tau_a \tau_b \otimes \tau_a \tau_b \int d^2 z_{12} \; e^{i\,(p,z_{12})} \times \frac{1}{|z_{12}|^2} \, . \tag{4.6}$$

Including the second diagram in Fig. 6, for the total prefactor we get

$$\frac{1}{\bar{z}_{12}} \, \tau_a \tau_b \otimes \left( \frac{1}{z_{12}} \tau_a \tau_b + \frac{1}{z_{21}} \tau_b \tau_a \right) = \frac{1}{|z_{12}|^2} \, \tau_a \tau_b \otimes [\tau_a, \tau_b] = -\frac{\mathsf{C}_2}{|z_{12}|^2} \, \tau_a \otimes \tau_a \, , \tag{4.7}$$

where $\mathsf{C}_2$ is defined by $\frac{1}{2} f_{abc} f_{abd} = \mathsf{C}_2 \delta_{cd}$, i.e. it is the value of the Casimir operator in adjoint representation. The one-loop contribution to the four-point function is thus

$$-\frac{1}{2\pi} \, \widehat{G}_4^{\text{1-loop}}(p) = \mathsf{C}_2 \, \frac{\varkappa^2}{2\pi} \int d^2 z_{12} \, \frac{1}{|z_{12}|^2} \times e^{i\,(p,z_{12})} := \varkappa^2 \, \mathsf{A}(p) \tag{4.8}$$

Notice that the presence of the exponential factor makes the integral IR-convergent (i.e. for $|z_{12}| \to \infty$) if $p \neq 0$, which is the reason we have kept the dependence on $p$. On the other hand, the integral is divergent in the UV. The divergence can be regularized in a variety of ways, all of which involve the introduction of a cutoff momentum scale $\Lambda$. For example, one can insert a factor if $(\Lambda|z_{12}|)^\epsilon$ in the integrand, where $\epsilon > 0$ is a

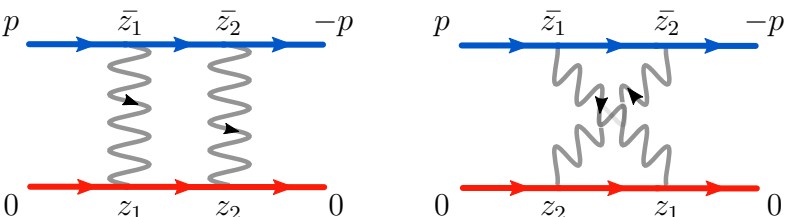

Figure 6: Diagrams contributing at one loop.

small positive number, or set a hard cutoff $|z_{12}| > \frac{1}{\Lambda}$. At higher loops we require that the regularization is multiplicative in the $z_{i\,i+1}$-variables. The one-loop $\beta$-function is independent of this regularization, as the following argument suggests. Instead of considering the actual integral (4.8), we calculate its derivative w.r.t. momentum, $p\frac{\partial \mathsf{A}(p)}{\partial p}$. The potential divergence is logarithmic, therefore proportional to $\log\left(\frac{p^2}{\Lambda^2}\right)$, and the derivative allows finding the coefficient of this divergence while dealing with a *convergent* integral. We find:

$$p\frac{\partial \mathsf{A}(p)}{\partial p} = \mathsf{C}_2 \int \frac{d^2 z_{12}}{2\pi}\,\frac{i\,p}{\bar{z}_{12}}\times e^{i\,(p,z_{12})} = -\frac{1}{2}\mathsf{C}_2\,, \tag{4.9}$$

so that, whatever the regularization might be, in the limit $\Lambda \to \infty$ one has

$$\mathsf{A}(p) = -\frac{1}{2}\mathsf{C}_2\,\log\left(\frac{p^2}{\Lambda^2}\right) + \text{const.} \tag{4.10}$$

As we shall see, the constant amounts to a finite renormalization of the coupling. One should also bear in mind that $\mathsf{C}_2 = n$ in the case of $\mathfrak{sl}_n$, so that effectively $\mathsf{A}(p) \sim n$.

**4.2. Two loops** The two-loop calculation can as well be performed directly and is rather instructive. Here we have $3! = 6$ diagrams, shown in Fig. 7. The corresponding value of the integrand is

$$\frac{\varkappa^3}{2\pi}\,e^{i\,(p,z_{13})}\times \tau_a\tau_b\tau_c\,\frac{1}{\bar{z}_{12}\bar{z}_{23}}\otimes\left(\tau_a\tau_b\tau_c\,\frac{1}{z_{12}z_{23}} + \boxed{\tau_a\tau_c\tau_b\,\frac{1}{z_{13}z_{32}}} + \tau_b\tau_a\tau_c\,\frac{1}{z_{21}z_{13}} + \right. \tag{4.11}$$
$$\left. +\tau_c\tau_a\tau_b\,\frac{1}{z_{31}z_{12}} + \boxed{\tau_b\tau_c\tau_a\,\frac{1}{z_{23}z_{31}}} + \tau_c\tau_b\tau_a\,\frac{1}{z_{32}z_{21}}\right)$$

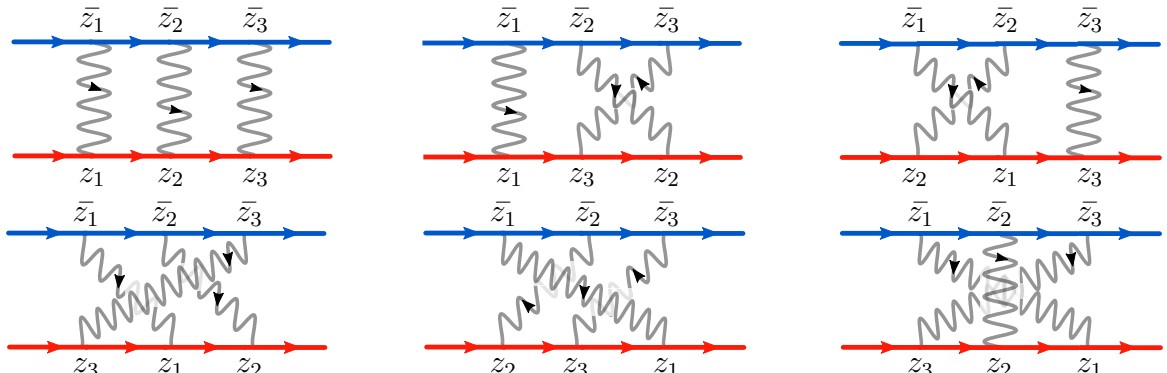

Figure 7: Diagrams contributing at two loops.

In the two terms in boxes we use the identity $\frac{1}{z_{13}z_{23}} = \frac{1}{z_{12}}\left(\frac{1}{z_{23}} - \frac{1}{z_{13}}\right)$. Regrouping the terms, we may rewrite this as

$$\frac{\varkappa^3}{2\pi} e^{i\,(p,z_{13})} \times \tau_a \tau_b \tau_c \frac{1}{\bar{z}_{12}\bar{z}_{23}} \otimes \left(\frac{1}{z_{12}z_{23}}[\tau_a,[\tau_b,\tau_c]] - \frac{1}{z_{12}z_{13}}[\tau_b,[\tau_a,\tau_c]]\right) \tag{4.12}$$

In the second term one has $\tau_a \tau_b \tau_c \otimes [\tau_b,[\tau_a,\tau_c]] = -\mathsf{C}_2\,\{\tau_a,\tau_b\} \otimes [\tau_a,\tau_b] = 0$. Simplifying the first term, we get

$$\frac{\varkappa^3}{2\pi} \frac{e^{i\,(p,z_{13})}}{|z_{12}|^2|z_{23}|^2} \tau_a\tau_b\tau_c \otimes [\tau_a,[\tau_b,\tau_c] = \frac{\varkappa^3}{2\pi} (\mathsf{C}_2)^2 \frac{e^{i\,(p,z_{13})}}{|z_{12}|^2|z_{23}|^2} \tau_a \otimes \tau_a. \tag{4.13}$$

What remains is to perform the integrals over $z_{12}$ and $z_{23}$. One point to notice is that the exponent $e^{i\,(p,z_{12})}$ of the one-loop expression (4.8) is now replaced by $e^{i\,(p,z_{13})}$. Just as in the one-loop case, it provides an effective infrared cutoff. As for a UV regularization, we will assume it is the same as in the one-loop case, but again we will not specify it. The result of the integration is

$$-\frac{1}{2\pi} \widehat{G}_4^{\text{2-loop}}(p) = \varkappa^3 \left(\mathsf{A}(p)\right)^2. \tag{4.14}$$

## 5.  Further loops

The crucial fact about (4.14) is that it is the square of the one-loop result (4.8) (up to an overall factor of $\varkappa$), as was first observed a long time ago in [Des88]. In that same paper it was contemplated that the divergences of crossed-ladder diagrams, at

any order, could be powers of the one-loop divergence, and their sum would therefore result in a geometric series. This is equivalent to the $\beta$-function being one-loop exact. Indeed, the solution of the one-loop RG equation

$$\frac{d\varkappa}{d \log \mu} = -\mathsf{C}_2 \varkappa^2 \tag{5.1}$$

is a geometric series:

$$\varkappa(\mu) = \frac{\varkappa_0}{1 + \mathsf{C}_2\varkappa_0 \, \log\left(\frac{\mu}{\mu_0}\right)} \, . \tag{5.2}$$

Below we shall find that, although at three loops the structure of the geometric series persists, at four loops one inevitably runs into an ambiguity corresponding to the choice of regularization/renormalization scheme. This does not invalidate the claim about the all-loop $\beta$-function, but rather shifts the goal towards finding the 'canonical' renormalization prescription in the realm of cGN models.

By extrapolation from Figs. 6-7 it is clear what the contributions to the four point function at higher loops are (at least for the given simplified kinematics). At $k - 1$ loops there are $k$ vertices on each line, and the diagrams correspond to the $k!$ possible contractions of the vertices on the upper line with the vertices on the lower line. The corresponding contribution to the integrand is

$$\mathcal{I}_{k-1} := \frac{\varkappa^k}{(2\pi)^{k-2}} \frac{e^{i(p,z_{1k})}}{\overline{z_{12}}\,\overline{z_{23}} \cdots \overline{z_{k-1k}}} \sum_{a_1,\dots,a_k} \sum_{p \in S_k} \frac{\tau^{a_1} \cdots \tau^{a_k} \otimes \tau^{a_{p(1)}} \cdots \tau^{a_{p(k)}}}{z_{p(1)p(2)}\,z_{p(2)p(3)} \cdots z_{p(k-1)p(k)}} \tag{5.3}$$

The product of $\overline{z_{ij}}$ factors at the front is a universal term arising from the contractions in the upper line of the diagram, which is a generalization of the expressions at the front of (4.7) and (4.11). The inner sum is over the $k!$ permutations: note that one permutes the generators $\tau_a$ and simultaneously the points $z_i$ in the lower line.

It turns out that one can rewrite the sum in (5.3) as

$$\mathcal{I}_{k-1} = \frac{\varkappa^k}{(2\pi)^{k-2}} \frac{e^{i(p,z_{1k})}}{\overline{z_{12}}\,\overline{z_{23}} \cdots \overline{z_{k-1k}}} \, \Phi_{k-1}(z_{12}, \cdots, z_{k-1k}) \times \sum_a \tau_a \otimes \tau_a \tag{5.4}$$

In other words, the tensor structure factorizes explicitly. In order not to dwelve in the technical details, we relegate the proof to Appendix A. As a result, we are only interested in the prefactor $\Phi_{k-1}$ defined in (5.4). From (4.7) and (4.13) we easily extract the one- and two-loop values:

$$\Phi_1(z_{12}) = -\mathsf{C}_2 \frac{1}{z_{12}}, \qquad \Phi_2(z_{12}, z_{23}) = (\mathsf{C}_2)^2 \frac{1}{z_{12}z_{23}}. \tag{5.5}$$

In general, $\Phi_k$ is obtained from the sum in (5.3). Calculating it gets more tedious with each loop, and starting from three loops we use Mathematica to accomplish this task. Henceforth we will restrict to the case of $\mathfrak{sl}_n$, setting $\mathsf{C}_2 = n$. We note that, at $k$ loops, $\Phi_k$ is a polynomial in $n$ of degree $k$. Evaluating the sum for several values of $n$ allows determining the coefficients of this polynomial (which are functions of $z_{ii+1}$).

### 5.1. Three loops At three loops the result is:

$$\Phi_3(z_{12}, z_{23}, z_{34}) = -\frac{n^3}{z_{12}z_{23}z_{34}} + \frac{2n}{z_{13}z_{14}z_{24}} \tag{5.6}$$

Here $z_{13} = z_{12} + z_{23}$, $z_{24} = z_{23} + z_{34}$ and $z_{14} = z_{12} + z_{23} + z_{34}$. Clearly, the integral of the first term gives a cube of the one-loop result (4.8), which is the next expected term of the geometric series. One then needs to prove that the integral of the second term does not contribute to the $\beta$-function or, in other words, that it is *finite*. We will exploit the same tool as before, namely we differentiate the integral

$$I_3(p) = \frac{1}{(2\pi)^3} \int d^2 z_{12}\, d^2 z_{23}\, d^2 z_{34} \frac{1}{z_{12}z_{23}z_{34}z_{13}z_{14}z_{24}}\, e^{i(p,z_{14})}, \tag{5.7}$$

w.r.t. momentum, $p\frac{\partial I_3(p)}{p}$. Choosing $(z_{12}, z_{34}, z_{14})$ as a new set of variables and rescaling $z_{12}, z_{34}$ by $z_{14}$, we find that the integral $\frac{ip}{(2\pi)^2} \int d^2 z_{14} \frac{e^{i(p,z_{14})}}{z_{14}} = -\frac{1}{4\pi}$ factors out. As a result,

$$p\frac{\partial I_3(p)}{\partial p} = -\frac{1}{8\pi^2} \int \frac{d^2 z_{12}\, d^2 z_{34}}{z_{12}z_{34}(1 - z_{12} - z_{34})(1 - z_{34})(1 - z_{12})} \tag{5.8}$$

A direct calculation shows that the integral is zero (see Appendix B), so that $\frac{\partial I_3(p)}{\partial p} = 0$. This means that $I_3(p)$ is a constant, and there is no divergent term. This constant is, in general, regularization-dependent (as discussed above, various cutoffs on the $z_{i\,i+1}$ variables can be chosen).

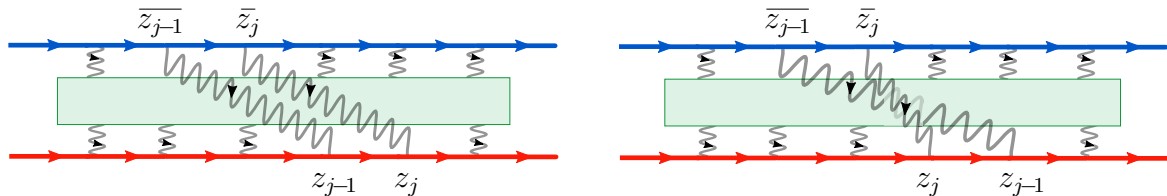

Figure 8: The two diagrams contributing to the residue in $z_{j-1j}$. The green box redistributes all other lines except the two at the front.

In any case, the integral of the second term in (5.6) is finite and, hence, it does not contribute to the $\beta$-function. As a result, at three loops one has

$$-\frac{1}{2\pi} G_4^{\text{3-loop}} = \varkappa^4 \left[ (\mathsf{A}(p))^3 - 2na \right],\tag{5.9}$$

where $a = I_3(p)$ is the regularization-dependent value of (5.7). In particular, the 3-loop divergence is again a power of the 1-loop result (4.8).

**5.2.  Isolating sub-divergences**  In passing to higher loops, an important step is the subtraction of subdivergences. Since so far we only encountered a divergence at one loop, let us discuss how it can be subtracted. A useful observation is that, at any loop order, the residues of $\Phi$ may be expressed through lower orders of perturbation theory:

$$\operatorname*{res}_{z_{j-1j}} \Phi_k(z_{12}, \cdots, z_{kk+1}) = -\mathsf{C}_2 \cdot \Phi_{k-1}(z_{12}, \cdots, \widehat{z_{j-1j}}, \cdots, z_{kk+1}),\tag{5.10}$$

where we normalize $\Phi_0 = 1$ and $\operatorname*{res}_{z_{j-1j}}$ means we are taking the residue at $z_{j-1j} = 0$.

To prove the statement, one should understand what diagrams lead to factors of $\frac{1}{z_{j-1j}}$ in $\Phi_k$. Such diagrams are exactly the ones, where the two gauge lines emanating from nodes $j - 1$ and $j$ in the top line run to adjacent nodes $s - 1$ and $s$ in the lower line, for any $s = 2, \cdots, k + 1$. This is shown in Fig. 8. For each $s$ there are exactly two such diagrams. Taking the residue means we drop the factor $\frac{1}{z_{j-1j}}$ and set $z_{j-1} = z_j$ in the rest of the diagram, which is tantamount to contracting the lines between the

$z_{j-1}$- and $z_j$-vertices. As for the color factor, for fixed $s$ one has

$$\sum_{a,b} \tau_{a_1} \cdots \tau_{a_{j-2}} \tau_a \tau_b \tau_{a_{j+1}} \cdots \tau_{a_k} \otimes \tau_{a_{p(1)}} \cdots \tau_{a_{p(s-2)}} (\tau_a \tau_b - \tau_b \tau_a) \tau_{a_{p(s+1)}} \cdots \tau_{a_{p(k)}} =$$

$$= -\mathsf{C}_2 \sum_a \tau_{a_1} \cdots \tau_{a_{j-2}} \tau_a \tau_{a_{j+1}} \cdots \tau_{a_k} \otimes \tau_{a_{p(1)}} \cdots \tau_{a_{p(s-2)}} \tau_a \tau_{a_{p(s+1)}} \cdots \tau_{a_{p(k)}} , \quad (5.11)$$

where the Casimir $\mathsf{C}_2$ appears due to the contraction of the structure constants $\frac{1}{2} \sum_{a,b} f_{abc} f_{abd} = \mathsf{C}_2 \delta_{cd}$, as before. Additionally, one has to multiply (5.11) by $z$-dependent factors, as in (5.3), and then sum over $s$, as well as over the permutations of the $k-2$ points $1, \cdots, j-2, j+1, \cdots k$. These two sums may be combined in a single sum over the permutations of $k-1$ points $1, \cdots, j-2, [j-1, j], j+1, \cdots k$, where $[j-1, j]$ is the 'merged point'. In other words, the color structure of the residue is exactly the same as one would have upon contracting the $z_{j-1}$- and $z_j$-vertices into a single vertex, up to an overall constant $-\mathsf{C}_2$. Effectively this is the same calculation as in the one-loop example (4.7).

To see how (5.10) works, one can apply it at the one-, two- and three-loop level, using (5.5) and (5.6).

**5.3. Four loops** At the three-loop level, formula (5.6), all poles are contained in the first term: we may write $\Phi_3 = -\frac{n^3}{z_{12} z_{23} z_{34}} + \widehat{\Phi}_3 (z_{12}, z_{23}, z_{34})$, where $\widehat{\Phi}_3$ is the three-loop result after subtraction, i.e. without poles in either of the variables $z_{12}$, $z_{23}$, or $z_{34}$.

We can now determine the structure of the poles at the four-loop level, using (5.10):

$$\Phi_4 = \frac{n^4}{z_{12} z_{23} z_{34} z_{45}} - \left( \frac{n}{z_{12}} \widehat{\Phi}_3 (z_{23}, z_{34}, z_{45}) + \frac{n}{z_{23}} \widehat{\Phi}_3 (z_{12}, z_{34}, z_{45}) + \right. \quad (5.12)$$

$$\left. \frac{n}{z_{34}} \widehat{\Phi}_3 (z_{12}, z_{23}, z_{45}) + \frac{n}{z_{45}} \widehat{\Phi}_3 (z_{12}, z_{23}, z_{34}) \right) + \widehat{\Phi}_4 (z_{12}, z_{23}, z_{34}, z_{45}) ,$$

where again $\widehat{\Phi}_4$ is what remains after one-loop subdivergences have been subtracted. This remnant is characterized by the fact that its residues at $z_{i\,i+1} = 0$ vanish. Taking the residue of $\Phi_4$, say, w.r.t. $z_{45}$, one gets $-n \cdot \Phi_3(z_{12}, z_{23}, z_{34})$, as required by (5.10). The subtraction (5.12) can be easily understood if one recalls the expression for the

four-point function to this order:

$$-\frac{1}{2\pi}\,\widehat{G}_4(p) = \varkappa + \varkappa^2 \mathsf{A}(p) + \varkappa^3 (\mathsf{A}(p))^2 + \varkappa^4 \left[(\mathsf{A}(p))^3 - 2na\right] + \tag{5.13}$$
$$+ \varkappa^5 \left[(\mathsf{A}(p))^4 - 4 \cdot 2na \cdot \mathsf{A}(p) + I_4(p)\right] + \dots$$

The three terms in the last bracket correspond to the three terms in (5.12). In particular, $I_4(p)$ is the four-loop analogue of the remainder integral (5.7):

$$I_4(p) = \frac{1}{(2\pi)^4} \int d^2 z_{12}\, d^2 z_{23}\, d^2 z_{34}\, d^2 z_{45}\, \frac{e^{i(p,z_{15})}}{z_{12} z_{23} z_{34} z_{45}} \cdot \widehat{\Phi}_4\left(z_{12}, z_{23}, z_{34}, z_{45}\right) \tag{5.14}$$

In (5.13) $\varkappa$ is the bare coupling constant. The geometric series in $\varkappa$ can be resummed as follows:

$$-\frac{1}{2\pi}\,\widehat{G}_4(p) = \varkappa(p) - \varkappa(p)^4\, 2na + \varkappa(p)^5\, I_4(p) + \dots\,, \tag{5.15}$$

$$\text{where} \qquad \varkappa(p) = \frac{\varkappa}{1 - \varkappa\,\mathsf{A}(p)} \tag{5.16}$$

One sees that the terms in (5.12) containing poles have been effectively interpreted in terms of lower orders of perturbation theory. Recalling the expression (4.10) for $\mathsf{A}(p)$, we introduce the renormalized coupling $\widetilde{\varkappa}$:

$$\frac{1}{\widetilde{\varkappa}} = \frac{1}{\varkappa} + \mathsf{C}_2 \log\left(\frac{\mu}{\Lambda}\right) - \text{const.} \tag{5.17}$$

Subtraction of the constant may be interpreted as a finite redefinition of the coupling. It is natural to subtract it together with the logarithm to ensure that there is no contribution to the $\beta$-function at three loops (one can think of this as a 'modified minimal subtraction'). The running coupling $\varkappa(p)$ is thus

$$\varkappa(p) = \frac{\widetilde{\varkappa}}{1 + \widetilde{\varkappa}\,\mathsf{C}_2 \log\left(\frac{|p|}{\mu}\right)} \tag{5.18}$$

We will apply to the integral $I_4(p)$ the technique used in the three-loop case, i.e.,

we differentiate it w.r.t. the momentum. The result of an explicit calculation is[11]:

$$p\frac{\partial I_4(p)}{\partial p} = n^2\left[-2a + \frac{6p\,i}{(2\pi)^4}\int\frac{d^2z_{12}d^2z_{23}d^2z_{34}d^2z_{45}}{\overline{z}_{12}\overline{z}_{23}\overline{z}_{34}\overline{z}_{45}}\left(\frac{1}{z_{13}z_{24}z_{25}} + \frac{1}{z_{13}z_{14}z_{25}}\right)e^{i(p,z_{15})}\right],$$
(5.19)

where $a$ is again the regularization-dependent (but finite and momentum-independent) value of the integral (5.7). In Appendix B we show that the remaining integral is $\frac{1}{p}$ times a constant that can be computed explicitly. The result is:

$$p\frac{\partial I_4(p)}{\partial p} = -2n^2\left[a + \frac{9\zeta(3)}{8}\right].$$
(5.20)

Thus, the four-loop contribution to the $\beta$-function in a 'minimal subtraction scheme' may be written as

$$\beta(\widetilde{\varkappa}) = -n\,\widetilde{\varkappa}^2 - 4n^2\left[a + \frac{9\zeta(3)}{8}\right]\widetilde{\varkappa}^5 + \dots$$
(5.21)

If one allows redefinitions of the coupling constant, as in (2.6), one may tune the parameter $c$ in such a way as to set the four-loop correction to zero. One might wonder, though, whether the $\zeta(3)$-piece in (5.21) has an invariant meaning. It turns out that it is related to the value of the $\beta$-function in the so-called momentum subtraction scheme.

**5.4.   Momentum subtraction (MOM) vs. $\overline{\text{MS}}$-scheme**   One could, of course, simply use the freedom in the (finite) redefinition of the coupling constant to cancel the ambiguous parameter $a$ in (5.21) (equal to the value of the integral $I_3(p)$ in a chosen regularization). This can be done at a more conceptual level, by defining the coupling constant as the value of the four-point function at a given value of momentum:

$$-\frac{1}{2\pi}\left.\widehat{G}_4\right|_{p^2=\mu^2} \equiv \varkappa_R.$$
(5.22)

This is a variation of the so-called momentum subtraction (MOM) scheme [CG79; BL81; CR00]. By definition, the $\beta$-function of $\varkappa_R$ is

$$\widehat{\beta}(\varkappa_R) := -\frac{1}{2\pi}\frac{\partial\widehat{G}_4(p)}{\partial\log|p|}\Big|_{p^2=\mu^2} = -\frac{1}{2\pi}2p\cdot\frac{\partial\widehat{G}_4(p)}{\partial p}\Big|_{p^2=\mu^2}$$
(5.23)

---

[11]Technically the expression for $I_4(p)$ obtained from the definitions (5.4), (5.12), (5.14) involves more terms, but many of them may be shown to be equal by permuting the variables $z_{12},\dots,z_{45}$ in the integrals.

Using expression (5.15) for $\widehat{G}_4$, we find

$$\widehat{\beta}(\varkappa_R) = \left[ 2p \left( -\frac{1}{2\pi} \frac{\partial \widehat{G}_4(p)}{\partial \varkappa(p)} \right) \frac{\partial \varkappa(p)}{\partial p} + 2p \cdot \varkappa(p)^5 \frac{\partial I_4(p)}{\partial p} + \dots \right]_{p^2 = \mu^2} = \quad (5.24)$$

$$= \left( 1 - 8na\,\varkappa(\mu)^3 + \dots \right) \cdot \left( -n\varkappa(\mu)^2 \right) + \varkappa(\mu)^5 \left( -4n^2 \right) \left[ a + \frac{9\zeta(3)}{8} \right] + \cdots,$$

where in passing to the second line we inserted the values of various derivatives from (5.15), (5.16) and (5.20). Taking into account that (to the relevant order) $\varkappa_R = \varkappa(\mu) - \varkappa(\mu)^4\, 2na + \dots$, we may express the $\beta$-function in the MOM-scheme in terms of $\varkappa_R$:

$$\widehat{\beta}(\varkappa_R) = -n\,\varkappa_R^2 - \frac{9}{2} n^2 \zeta(3)\,\varkappa_R^5 + \dots \qquad (5.25)$$

The fact that the regularization-dependent constant $a$ drops out is compatible with the 'regularization-independent' property of the MOM-scheme known in the literature [CG79; KS14a].

It follows from (5.25) that the scheme defined by (5.22) violates the one-loop-exactness of the $\beta$-function. In 4D SUSY theories it has also been observed that the NSVZ relation is violated in the MOM scheme [KS14b]. Here the relevant counterpart is SUSY electrodynamics with $N_f$ flavors of electrons, since the $\mathbb{CP}^{n-1}$ model is as well an abelian theory with $N_f = n$ matter fields. It was found that the $\beta$-function of SQED features a $\zeta(3)$-term in the MOM scheme, which disappears in other schemes such as $\overline{\text{MS}}$. One simple reason why the MOM scheme is not optimal is that it depends on the configuration of momenta in the definition of the coupling constant (5.22). The configuration we have chosen is technically simple, but it does not seem to be in any sense 'natural'. One would prefer to set the external momenta equal to zero, however in that case one encounters an IR divergence, as already discussed.

One might then be tempted to switch to the $\overline{\text{MS}}$ scheme, where, according to (3.1), the corresponding contribution to the $\beta$-function should vanish (on the sigma model side). Here a technical problem arises, since in $d = 2 - \epsilon$ dimensions the Gross-Neveu model is not multiplicatively renormalizable [Bon$^+$90; VDK95; VV97] due to the emergence of the so-called evanescent operators in perturbation theory (these are operators of the type $\left( \overline{\psi}[[\gamma^\alpha, \gamma^\beta] \cdots ]\psi \right)^2$ that may be defined formally and are non-zero

for $d \neq 2$). Interestingly the onset of the effects related to such operators, which make the calculation of the $\beta$-function substantially more complicated, is exactly at four loops, cf. [Gra08]. Even if the technical difficulties related to evanescent operators are overcome, one should bear in mind that the $\overline{\text{MS}}$ scheme itself has crucial drawbacks. For example, it breaks SUSY at a sufficiently high loop order [AV83]. In models with $\mathcal{N} = 1$ SUSY in 4D, this scheme breaks the NSVZ relation at three loops [JJN96; JJN97], which is the first order where scheme dependence appears. In the case of extended SUSY ($\mathcal{N} = 2$ in 4D, or $\mathcal{N} = (2,2)$ in 2D, as in our case) the mismatch might be postponed to higher loops. To summarize, there seems to be no reason to expect that the $\overline{\text{MS}}$ scheme should lead to a complete proof of the exact $\beta$-function (in theories with or without SUSY).

In four-dimensional theories a renormalization scheme applicable at *any* loop order has been found (cf. [KS13; SS22; KS14b; KS14a] and references therein) – this is the so-called higher covariant derivative method [Sla71; Sla72], supplemented by a minimal subtraction of logarithms. Perhaps this method could be adapted to our sigma model/Gross-Neveu setup.

## 6. Conclusion and outlook

In the present paper we studied quantum aspects of chiral Gross-Neveu models, most importantly their higher-loop $\beta$-function. These theories are especially interesting due to the recently discovered exact equivalence to certain sigma models, such as the well-known $\mathbb{CP}^{n-1}$ model[12]. This equivalence holds at the quantum level and implies, in particular, that the $\beta$-functions of the theories should coincide. The all-loop $\beta$-functions for very general classes of cGN models (including their deformations etc.) have been proposed in [GLM01], using ideas from the early work [Kut89]. If these match with sigma model calculations, they would provide the $\beta$-functions of a broad class of sigma models, including potentially their trigonometric and elliptic deformations[13].

One stumbling point along this promising path is a four-loop discrepancy between

---

[12]Generalizations to Grassmannians and other target spaces may be considered in a similar fashion.

[13]One could think of the 'sausage model' [FOZ93] as a prominent example. In this special case there is also a 'geometric' conjecture for its all-loop $\beta$-function in [HLT19].

a direct calculation and the proposed $\beta$-functions in a special case ('level-zero', $k = 0$), reported in [LW03]. As we explained above, this discrepancy might be attributed to a choice of renormalization scheme. In fact, the similar story of the NSVZ $\beta$-function (cf. [Shi12]) suggests that finding an explicit scheme where the $\beta$-function is of the conjectured form might be a complicated task. Amusingly, the equivalence between cGN models and sigma models mentioned earlier provides an important hint. It turns out that, via this equivalence, one way of explicitly realizing the special 'level-zero' case of [LW03] is by considering the SUSY $\mathbb{CP}^{n-1}$ sigma model. As opposed to the case of more general models, the $\beta$-function of this model is well studied and is known to be one-loop exact, if one uses a manifestly supersymmetric renormalization prescription. This is consistent with the general proposal of [GLM01] in this special case.

On the other hand, there are several reasons why one would *not* want to rely on explicit supersymmetry. First of all, the cGN formulation is not manifestly supersymmetric, and constructing a SUSY-compatible regularization seems a complicated task. Besides, the exact $\beta$-functions proposed in [GLM01] should be applicable in a much wider context (in particular, to non-SUSY models), and one would like to eventually verify the conjectures in those cases as well. Having this in mind, in the present paper we have carried out an explicit calculation of the $\beta$-function in the $k = 0$ (level-zero) cGN model in a version of momentum-subtraction (MOM) scheme. In this scheme the coupling constant is defined as the value of the four-point function for special value of momenta. We have shown that, although at the three-loop level the $\beta$-function is zero, at four loops it acquires a correction proportional to $n^2\zeta(3)$, which, again, could be eliminated by a coupling constant redefinition. One can draw a parallel with $\mathcal{N} = 1$ SQED in 4D, where an analogous MOM scheme leads to a $\zeta(3)$ contribution to the $\beta$-function at three loops (incompatible with the NSVZ expression) [KS14b].

It would be interesting and important to find a natural renormalization scheme where the corrections at four and higher loops to the level-zero cGN model $\beta$-function vanish. The special case of $k = 0$ seems to be the most 'purified' one from the calculational perspective, which is why it especially deserves further study. Once this is done, one would try to understand, whether, using similar methods, one can obtain all-loop $\beta$-functions for many other sigma models, such as the $\mathbb{CP}^{n-1}$ model with minimally coupled fermions, Grassmannian models, their trigonometric and elliptic deformations. If so, they could imply nontrivial physical consequences for the phase structure of these

theories. The interrelation of these exact results with the conjectured integrability of the models is another question that is worth being studied in detail.

**Acknowledgments.** This work is supported by the Russian Science Foundation grant № 22-72-10122 and by the Foundation for the Advancement of Theoretical Physics and Mathematics "BASIS". I would like to thank S. Derkachov, A. Pribytok, A. Smilga, K. Stepanyantz, A. Yung for discussions.

# Appendix A. Tensor structure of the four-point function

Here we prove that the sum (5.3), which defines the four-point function, has the tensor structure shown in (5.4).

The idea of the proof is in taking residues w.r.t. $z_1, \ldots, z_{k-1}$. We write

$$\mathcal{I}_{k-1} = \frac{e^{i(p,z_{1k})}}{\bar{z}_{12}\bar{z}_{23}\cdots\bar{z}_{k-1k}} \; Q_{k-1}(z_{12},\cdots,z_{k-1k}), \tag{A.1}$$

with $Q_{k-1}$ a meromorphic function. We may decompose this function in $z_1$-poles: $Q_{k-1} = \sum_{i=2}^{k} \frac{1}{z_{1i}} Q^{(i)}(z_{23},\cdots,z_{k-1k})$. Then we decompose in $z_2$-poles etc., finally arriving at

$$Q_{k-1} = \sum_{i_1\geqslant 2,\, i_2\geqslant 3,\, \cdots,\, i_{k-2}\geqslant k-1} \frac{1}{z_{1\,i_1}}\frac{1}{z_{2\,i_2}}\cdots\frac{1}{z_{k-1\,k}} \times Q^{i_1,\ldots,i_{k-2},k}, \tag{A.2}$$

where $Q^{i_1,\ldots,i_{k-2},k}$ encodes the tensor structure and does not depend on $z_{i\,i+1}$. Let us study what this tensor structure is.

When taking the residue of $Q_{k-1}$ w.r.t. $z_{1i}$, the only terms in the sum (5.3) that contribute are the ones with $p(j) = 1, p(j+1) = i$ or $p(j) = i, p(j+1) = 1$ for some $j$ (just like in Fig. 8 before, which corresponds to the special case $i = 2$). The sum of these two terms (which come with opposite signs) produce a commutator $[\tau^{a_1}, \tau^{a_i}]$ in position $j$ in the numerator. Denoting by $p' \in S_{k-2}$ permutations of the remaining

variables, i.e. of the set $\{1, \ldots, j-1, j+2, \ldots k\}$ onto $\{2, \ldots, i-1, i+1, \ldots, k\}$, we get:

$$\underset{z_{1i}}{\mathsf{res}} \sum_{p \in S_k} \frac{\tau^{a_{p(1)}} \cdots \tau^{a_{p(k)}}}{z_{p(1)p(2)} z_{p(2)p(3)} \cdots z_{p(k-1)p(k)}} = \sum_{p=p', \, j} \frac{\tau^{a_{p(1)}} \cdots \overset{\text{position } j}{\overbrace{\left[\tau^{a_1}, \tau^{a_i}\right]}} \cdots \tau^{a_{p(k)}}}{z_{p(1)p(2)} \cdots z_{p(j-1) \, i} z_{i \, p(j+2)} \cdots z_{p(k-1)p(k)}} =$$

$$= \sum_{p \in S_{k-1}} \frac{\widehat{\tau}^{a_{p(1)}} \cdots \widehat{\tau}^{a_{p(k)}}}{z_{p(1)p(2)} z_{p(2)p(3)} \cdots z_{p(k-2)p(k-1)}}, \tag{A.3}$$

where $\widehat{\tau}^{a_j} = \tau^{a_j}$ for $j \neq i$ and $\widehat{\tau}^{a_i} = \left[\tau^{a_1}, \tau^{a_i}\right]$. In other words, we obtain a sum similar to the original one, albeit with a $k \to k-1$ reduction and a redefinition of $\tau$'s. At the next step, i.e. upon taking a residue w.r.t. $z_{2i_2}$, we again obtain a similar sum with new variables $\widehat{\widehat{\tau}}^{a_j} = \widehat{\tau}^{a_j}$ for $j \neq i_2$, and $\widehat{\widehat{\tau}}^{a_{i_2}} = \left[\widehat{\tau}^{a_2}, \widehat{\tau}^{a_{i_2}}\right]$.

Upon iterating this procedure, we find that $Q^{i_1, \ldots, i_{k-1}}$ is a sum of nested commutators, which, upon expanding these commutators, may be simplified to $\sum_a S_a \otimes \tau^a$. By symmetry of the construction, we have $S_a = b\,\tau^a$ for constant $b$. We thus find that $Q_{k-1}$ is proportional to $\sum_a \tau^a \otimes \tau^a$. ............................................■

## Appendix B. Calculation of integrals

In this Appendix we compute the finite integrals encountered in the main text. The 'master' integral that we will be exploiting (as well as its complex conjugate) is

$$\int \frac{d^2 v}{\overline{v}(v+a)(v+b)} = \frac{\pi}{a-b} \log\left(\frac{|a|^2}{|b|^2}\right). \tag{B.1}$$

**Three-loop integral**. We start with the integral (5.8) entering at three loops:

$$p \frac{\partial I_3(p)}{\partial p} = -\frac{1}{8\pi^2} \int \frac{d^2 z_{12} \, d^2 z_{34}}{\overline{z_{12}} \overline{z_{34}}(1 - \overline{z_{12}} - \overline{z_{34}})(1 - z_{34})(1 - z_{12})} \tag{B.2}$$

Applying (B.1) to the $z_{34}$-integral, we get

$$p \frac{\partial I_3(p)}{\partial p} = \frac{1}{8\pi} \int \frac{d^2 z_{12} \, \log|z_{12}|^2}{\overline{z_{12}} \, |1 - z_{12}|^2}. \tag{B.3}$$

We split this integral in two regions: $|z_{12}| < 1$ and $|z_{12}| > 1$. Changing variables $z_{12} \to \frac{1}{z_{12}}$ in the second region, we find that the two integrals are equal but have opposite sign, so that $\frac{\partial I_3(p)}{\partial p} = 0$.

**First integral in (5.19).** We pass over to the four-loop integrals entering (5.19). The first one is

$$i_1 = \frac{i\,p}{(2\pi)^4} \int \frac{d^2 z_{12} d^2 z_{23} d^2 z_{34} d^2 z_{45}}{\bar{z}_{12}\bar{z}_{23}\bar{z}_{34}\bar{z}_{45}} \frac{1}{z_{13}z_{24}z_{25}} e^{i(p,z_{15})} \tag{B.4}$$

We pass to a new set of variables $v_1 = z_{12}, v_2 = z_{23}, v_3 = z_{34}$ and $v := z_{15}$. Rescaling $v_1, v_2, v_3$ by $v$, we find that the $v$-integral factors out:

$$i_1 = \frac{i\,p}{(2\pi)^4} \int d^2 v\, \frac{e^{i(p,v)}}{\bar{v}} \cdot \int \frac{d^2 v_1\, d^2 v_2\, d^2 v_3}{\bar{v}_1 \bar{v}_2 \bar{v}_3 (1 - \bar{v}_1 - \bar{v}_2 - \bar{v}_3)} \cdot \frac{1}{(v_1 + v_2)(v_2 + v_3)(1 - v_1)} \tag{B.5}$$

To proceed, we apply (B.1) to the $v_3$-integral:

$$i_1 = -\frac{1}{16\pi^2} \int \frac{d^2 v_1\, d^2 v_2}{\bar{v}_1 \bar{v}_2 (1 - \bar{v}_1 - \bar{v}_2)} \cdot \frac{1}{(v_1 + v_2)(1 - v_1)} \log\left(\frac{|1 - v_1|^2}{|v_2|^2}\right) \tag{B.6}$$

Shifting $v_1 \to 1 - v_1$ and then rescaling $v_2 \to v_1 v_2$, we again arrive at a $v_1$-integral that can be calculated using (B.1):

$$i_1 = -\frac{1}{16\pi} \int d^2 v_2\, \frac{(\log(|v_2|^2))^2}{\bar{v}_2 |1 - v_2|^2} \tag{B.7}$$

We split this integral in two: $|v_2| < 1$ and $|v_2| > 1$. Making in the second one the change of variables $v_2 \to \frac{1}{v_2}$ and simplifying slightly, we arrive at

$$i_1 = -\frac{1}{8\pi} \int_{|v_2|<1} d^2 v_2\, \frac{(\log(|v_2|^2))^2}{1 - |v_2|^2} = -\frac{\zeta(3)}{4}\,. \tag{B.8}$$

**Second integral in (5.19).** Next we switch to the second integral in (5.19):

$$i_2 = \frac{i\,p}{(2\pi)^4} \int \frac{d^2 z_{12} d^2 z_{23} d^2 z_{34} d^2 z_{45}}{\bar{z}_{12}\bar{z}_{23}\bar{z}_{34}\bar{z}_{45}} \frac{1}{z_{13}z_{14}z_{25}} e^{i(p,z_{15})} \tag{B.9}$$

Using the same variables as before, one can perform the integral over $z_{15} = v$:

$$i_2 = -\frac{1}{16\pi^3} \int \frac{d^2v_1\, d^2v_2\, d^2v_3}{\bar{v}_1 \bar{v}_2 \bar{v}_3 (1 - \bar{v}_1 - \bar{v}_2 - \bar{v}_3)} \cdot \frac{1}{(v_1 + v_2)(v_1 + v_2 + v_3)(1 - v_1)} \quad \text{(B.10)}$$

Using (B.1), we integrate over $v_3$ and shift $v_2 \to v_2 - v_1$:

$$i_2 = \frac{1}{16\pi^2} \int \frac{d^2v_1\, d^2v_2\, \log|v_2|^2}{\bar{v}_1(\bar{v}_2 - \bar{v}_1)(1 - \bar{v}_2)v_2(1 - v_1)} \quad \text{(B.11)}$$

We may now again employ (B.1) for integration over $v_1$. The resulting integral is split into $|v_2| < 1$ and $|v_2| > 1$, and in the second piece we switch $v_2 \to \frac{1}{v_2}$. After some simplifications we get

$$i_2 = -\frac{1}{8\pi} \int\limits_{|v_2| < 1} \frac{d^2v_2}{|v_2|^2} \log\left(|v_2|^2\right) \log\left(1 - |v_2|^2\right) = -\frac{\zeta(3)}{8} . \quad \text{(B.12)}$$

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
