# Peer review of "$β$-function of the level-zero Gross-Neveu model"

_SciPost Physics_

## Round 1 · Referee Report · Anonymous (Referee 1) · 2023-2-3

Report
The paper has a clear formulation of the problem and solution argumentation appears to be complete within the framework provided. It possesses very well established structure along with necessary derivations that makes the paper self-consistent overall.
The computation of observables in sigma models and their deformations is an important task, that interrelates several subjects of string and field theory. In particular, the renormalisability and all-loop exactness of $ \beta $-function in 2- and 4-dim supersymmetric gauge theories is a crucial long-standing problem. In the present work the author not only addresses the problem of $ \beta $-function one-loop exactness and its computation for $ k = 0 $ Gross-Neveu model, but provides a novel formalism for studying different types of observables arising in Sigma and Field theory models. The last is realised through the mapping relation of supersymmetric $ \mathbb{CP}^{n-1} $ model to chiral Gross-Neveu model, which allows to approach the problem by means of field theoretic apparatus and gain understanding on both sides of correspondence.
Additional useful extensions and comments might include (not affecting recommendation for publication):
In the computation of 4-loop $ \beta $-function, the MOM-scheme is always dependent on the configuration of momenta as of (5.22) (no matter if asymmetric or a different momenta configuration is chosen), which consequently leads to one-loop exactness breaking. Might it be possible to implement analogous scheme in other spaces, where $ \zeta(3) $ transcendentality would not emerge and supersymmetry is respected? Is it understood if in MOM-scheme $ \zeta $-dependence is universal or how does it develop at higher orders? [ Possibly, understanding of the last or its resummation would lead to consistent correlation with Gerganov-LeClair-Moriconi proposal. ]
In relation to similar properties of the 4-dim supersymmetric theories, e.g. $ \mathcal{N} = 1, 2 $ or SQED, where an analogy in transcendental dependence occurs. Are there other reasons for this to hold, i.e. apart from the fact that $ \mathbb{CP}^{n-1} $ is a theory with $ N_{f} $ matter degrees (eventually leading to similar class of integrands)? Likewise, are there physical reasons for corrections arising in $ \mathcal{N}=2 $ $ \beta_{\,\text{4-loop}}^{\,\text{NSVZ}} $ and $ \beta_{\,\text{4-loop}}^{\,\text{cGN}} $ ? Is it known if in 2-dim higher covariant derivatives (through Slavnov procedure) would solve regularisation/renormalisation dependence or problem would remain like in 4-dim case?
If it is possible to embed present formalism for systems where left/right propagating sector is absent (like $ (0,2) $-model), is it known whether GN type/Sigma model mapping would still hold for these classes? In this respect, since in GN formalism it is proper to consider phase space formulation, might analogous technique work for a system with orthosymplectic space (e.g. bow quiver variety)?
The work results are original and form important contribution in the field of integrable systems and Sigma models.
The computation of observables in sigma models and their deformations is an important task, that interrelates several subjects of string and field theory. In particular, the renormalisability and all-loop exactness of $ \beta $-function in 2- and 4-dim supersymmetric gauge theories is a crucial long-standing problem. In the present work the author not only addresses the problem of $ \beta $-function one-loop exactness and its computation for $ k = 0 $ Gross-Neveu model, but provides a novel formalism for studying different types of observables arising in Sigma and Field theory models. The last is realised through the mapping relation of supersymmetric $ \mathbb{CP}^{n-1} $ model to chiral Gross-Neveu model, which allows to approach the problem by means of field theoretic apparatus and gain understanding on both sides of correspondence.
Additional useful extensions and comments might include (not affecting recommendation for publication):
In the computation of 4-loop $ \beta $-function, the MOM-scheme is always dependent on the configuration of momenta as of (5.22) (no matter if asymmetric or a different momenta configuration is chosen), which consequently leads to one-loop exactness breaking. Might it be possible to implement analogous scheme in other spaces, where $ \zeta(3) $ transcendentality would not emerge and supersymmetry is respected? Is it understood if in MOM-scheme $ \zeta $-dependence is universal or how does it develop at higher orders? [ Possibly, understanding of the last or its resummation would lead to consistent correlation with Gerganov-LeClair-Moriconi proposal. ]
In relation to similar properties of the 4-dim supersymmetric theories, e.g. $ \mathcal{N} = 1, 2 $ or SQED, where an analogy in transcendental dependence occurs. Are there other reasons for this to hold, i.e. apart from the fact that $ \mathbb{CP}^{n-1} $ is a theory with $ N_{f} $ matter degrees (eventually leading to similar class of integrands)? Likewise, are there physical reasons for corrections arising in $ \mathcal{N}=2 $ $ \beta_{\,\text{4-loop}}^{\,\text{NSVZ}} $ and $ \beta_{\,\text{4-loop}}^{\,\text{cGN}} $ ? Is it known if in 2-dim higher covariant derivatives (through Slavnov procedure) would solve regularisation/renormalisation dependence or problem would remain like in 4-dim case?
If it is possible to embed present formalism for systems where left/right propagating sector is absent (like $ (0,2) $-model), is it known whether GN type/Sigma model mapping would still hold for these classes? In this respect, since in GN formalism it is proper to consider phase space formulation, might analogous technique work for a system with orthosymplectic space (e.g. bow quiver variety)?
The work results are original and form important contribution in the field of integrable systems and Sigma models.

---

## Round 1 · Referee Report · Anonymous (Referee 2) · 2023-2-21

Report
The paper is devoted to the investigation of quantum corrections in the chiral Gross-Neveu models and, in particular, the higher order contributions to the β-function. The author performs the explicit calculation of the four-loop β-function in the level-zero chiral Gross-Neveu model and compares the result with the earlier proposed exact expression. It is demonstrated that in momentum subtraction scheme the exact expression is not valid, but the disagreement can be attributed to the scheme dependence of the β-function. The difference of the four-loop result and the corresponding part of the exact expression is proportional to ζ(3). The author notes that this is analogous to the situation with the exact NSVZ β-function in D=4 N=1 supersymmetric electrodynamics, where such a disagreement arises in three loops and also appears due to the scheme dependence. The calculation made in the paper is technically complicated, its result seems very interesting, and the conclusions are quite reasonable. That is why I recommend to publish the paper in its present form.

---

## Editorial Decision

resubmitted